# Analog-Domain Suppression of Strong Interference Using Hybrid Antenna Array

**DOI:** 10.3390/s22062417

**Published:** 2022-03-21

**Authors:** Kai Wu, J. Andrew Zhang, Xiaojing Huang, Y. Jay Guo, Diep N. Nguyen, Asanka Kekirigoda, Kin-Ping Hui

**Affiliations:** 1Global Big Data Technology Centre (GBDTC), University of Technology Sydney (UTS), Sydney, NSW 2007, Australia; kai.wu@uts.edu.au (K.W.); andrew.zhang@uts.edu.au (J.A.Z.); xiaojing.huang@uts.edu.au (X.H.); diep.nguyen@uts.edu.au (D.N.N.); 2Defence Science and Technology (DST), Edinburgh, SA 5111, Australia; asanka.kekirigoda@dst.defence.gov.au (A.K.); kinping.hui@dst.defence.gov.au (K.-P.H.)

**Keywords:** interference suppression, hybrid antenna array, null steering, constant-modulus phase shifter, beamforming, angle-of-arrival (AoA) estimation

## Abstract

The proliferation of wireless applications, the ever-increasing spectrum crowdedness, as well as cell densification makes the issue of interference increasingly severe in many emerging wireless applications. Most interference management/mitigation methods in the literature are problem-specific and require some cooperation/coordination between different radio frequency systems. Aiming to seek a more versatile solution to counteracting strong interference, we resort to the hybrid array of analog subarrays and suppress interference in the analog domain so as to greatly reduce the required quantization bits of the analog-to-digital converters and their power consumption. To this end, we design a real-time algorithm to steer nulls towards the interference directions and maintain flat in non-interference directions, solely using constant-modulus phase shifters. To ensure sufficient null depth for interference suppression, we also develop a two-stage method for accurately estimating interference directions. The proposed solution can be applicable to most (if not all) wireless systems as neither training/reference signal nor cooperation/coordination is required. Extensive simulations show that more than 65 dB of suppression can be achieved for 3 spatially resolvable interference signals yet with random directions.

## 1. Introduction

Due to the proliferation of wireless systems, the ever-increasing spectrum crowdedness, as well as the densification of cellular networks, radio frequency (RF) interference has reappeared to be a major challenge in many emerging wireless applications [1,2,3,4,5,6,7,8]. This, for instance, happens in wirelessly powered communications (WPC) [2], where the range difference between power and information transfers makes the information signal weaker than the power transfer signal by many orders of magnitude [3]. Another example where the interference is highly detrimental can be found in the convergence of radar sensing and wireless communications that has attracted extensive attention recently [4,5,6]. Given its high transmission power, a radar can cause strong interference to the nearby communication system [7]. A further example showing the destructive impact of the strong interference is in the increasingly popular full-duplex communications; refer to [8] for details.

Numerous interference management/mitigation methods have been proposed. The resource scheduling is applied for relieving the mutual interference in WPC [9,10] and also in the co-existence of radar and communications (CRC) [11]. As pointed out in [2], this method can reduce spectral efficiency and puts strict requirement on time synchronization between different RF systems. Moreover, this method relies on the information exchange between participant RF systems [12]. Another popular method of suppressing RF interference is the null-space projection-based transmission, as applied in CRC [13,14] and full-duplex communications [15]. In particular, this method designs the transmitting beamforming/precoding to project transmitted signals onto the null-space of the interference matrix between the involved RF systems. Despite the effectiveness of the above methods, they require specially designed training signals between the participating RF systems for estimating the aforementioned inference matrix. In the case of massive MIMO systems, the training overhead can be heavy. To this end, it is desirable if a receiver itself has the capability of suppressing strong interference signals without requiring lengthy training from nearby transmitters or RF sources.

Antenna-array-based receivers can suppress interference signals by performing adaptive beamforming [16,17]. The well-known optimum criteria for adaptive beamforming include the minimum variance distortion-less response (also known as Capon), the maximum signal-to-interference-plus-noise ratio, and the linear constraints minimum variance [18]. These adaptive beamforming criteria generally rely on the fully digitized signal samples and hence require a high-dynamic-range front end and ADC to be equipped for each antenna. This requirement further leads to the high power consumption and cost of the digital-array-based receiver [19]. Moreover, given the limited dynamic range, information-bearing signals can be severely corrupted by clipping or quantization noises [2,20]. This makes the conventional adaptive beamforming-based interference suppression not sufficiently effective due to the potential loss/distortion of the direction information in the expected signals.

In contrast to the fully digital array, the hybrid antenna array of analog subarrays, as illustrated in Figure 1, has been recognized as a cost-effective and energy-efficient solution to counteracting strong RF interference [21,22]. This is attributed to its two significant benefits. One is that the analog subarrays can suppress strong interference signals prior to frequency conversion and digitization, hence easing the dynamic range requirements and reducing the power consumption of RF chains [23]. The other benefit of using a hybrid array is that the spatial degrees of freedom (DoF) are preserved given multiple analog subarrays. However, to enjoy these benefits of hybrid arrays, we need an efficient design of the analog beamformer that flexibly steers nulls towards strong interference signals while maintaining a flat amplitude response in the spatial passband, i.e., non-interference directions [2,23,24]. Hereafter, we refer to such a beamformer as the analog interference-nullifying beamformer (AINB).

So far, only a few works have specifically considered the AINB design in hybrid arrays. In [2], the AINB is constructed as a Kronecker product of component vectors, and each vector is designed based on the truncated Fourier/Hardamard matrices. However, this design only focuses on forming nulls, which may lead to severe attenuation in the spatial passband. In [24], the AINB is designed by successively approximating the desired beamformer as a linear combination of implementable analog beamformers. This method requires a reference signal which, however, would be fully corrupted by the strong quantization noises in the presence of interference signals, and the reference signal may not always be available. In [23], the AINB is achieved by optimizing the RF input impedance with the phase shifters used for shifting the null to the interference direction. Since only a single null is steered, this design can be incompetent in the presence of multiple interference signals.

To address the issues in the above works, we design a novel AINB by minimizing the spatial responses towards the interference directions and approximately maximizing the beam flatness in the spatial passband. Such a design can quickly adapt to the dynamic interference environment and form multiple nulls as required. However, due to the constant modulus constraints on the phase shifters in an analog subarray, the AINB design problem is highly non-convex and non-trivial to be solved efficiently in real time. Moreover, to steer nulls towards interference signals, the accurate estimations of the interference directions are required. In this paper, we consider the case where the power difference between interference and information-bearing signals is greater than the dynamic range of a receiver. In such a case, the AGCs in RF chains will reduce the receiving gains to prevent signal clipping, as illustrated in Figure 1. As a result, information-bearing signals will have too large quantization noises, which makes it infeasible to estimate the directions of interference signals. The AINB sequentially designed can substantially suppress interference signals, triggering the AGCs to adjust for the reception of information-bearing signals. The key contributions and novelties of this paper are summarized as follows.

We develop an efficient solver for designing the phase-only AINB under the framework of majorization–minimization (MM). In particular, we design the objective function in such a way that we are able to simplify it substantially based on the newly unveiled relation between the spatial responses in the interference directions and the spatial passband. Thanks to the simplification of the objective function, we then propose a low-complexity method of constructing the majorization function, where we manage to remove the need of computationally intensive eigenvalue decomposition (EVD) required in the conventional construction. We further derive an iterative solver for the AINB design, where a closed-form solution with low complexity is achieved in each iteration. In addition, the impact of the initial solution to the proposed solver is investigated, based on which a high-quality initialization for the solver is established;We develop a two-stage angle of arrival (AoA) estimation method based on the conventional ESPRIT (estimation of signal parameters via invariance techniques). A major innovation of the method is the design of subarray beamforming in the two stages. In particular, an omnidirectionally flat beam is produced at each subarray in the first stage, while in the second stage the beam is created towards each of the AoA estimates obtained previously. To the best of our knowledge, it has not been investigated in the literature to use specially optimized flat beams for improving the AoA estimation performance of ESPRIT in hybrid antenna arrays.We provide extensive simulation results to validate the effectiveness of the proposed designs. As for the AINB, we cannot find similar designs in the literature. Thus, we comprehensively evaluate and observe numerous performance metrics, including spatial responses, interference suppression capability, and convergence curves for designing AINBs over tens of thousands of independent trials. As for the AoA estimation method, we employ the state-of-the-art [25] as a benchmark for the reasons to be explained at the beginning of Section 5. Due to the proposed use of deliberately optimized flat beams, the AoA estimation performance is substantially improved over the prior art [25]. Moreover, thanks to the high accuracy of the proposed AoA estimation method, the proposed AINB design can efficiently steer deep nulls towards interference signals.

It should be pointed out that it is not the first effort to apply ESPRIT to the hybrid arrays. The state of the art, such as that in [25], does so by focusing on augmenting the signal subspace at the expense of estimation time. Our study was motivated to achieve a large dimension of the signal subspace in the hybrid array that is not limited to the number of RF chains. This, however, is not a severe issue in our case, since the number of strong interference signals that are required to be suppressed in the analog domain is generally small [23]. We also remark that the hybrid array, which is also called a massive MIMO arrays, has been extensively studied in the context of the millimeter-wave (mmWave) and also terahertz communications for 5G and beyond [26,27,28,29,30]. Given the sparse nature of the mmWave channels, the AoA estimation methods recently developed for hybrid arrays, e.g., [31,32,33], mainly target at the LoS-dominated scenarios. In contrast, our design here assumes several strong interference signals.

The remainder of the paper is organized as follows. Section 2 formulates the AINB design problem, along with the signal model established. Section 3 elaborates the MM-based algorithm for designing AINB, with the objective simplification, algorithm development, and initialization detailed in three subsections. Section 4 illustrates the proposed two-stage AoA estimation method. The simulation results are provided in Section 5, followed by Section 6, which concludes the paper.

## 2. Problem Formulation

As shown in Figure 1, we consider a uniform linear array of *M* analog subarrays. Each subarray, having *N* antennas, performs analog beamforming using phase shifters. The spacing between any adjacent antennas is identical, as denoted by *d*. Note that the antenna coupling can be very small when d≥λ/2 [21], where λ denotes the wavelength. Thus, to focus on introducing the proposed interference suppression scheme, we assume d≥λ/2 and ignore antenna coupling in this work. In the normal receiving array, the signals, impinging on the antennas, are first combined through subarray beamforming in the analog domain; then, each subarray output is processed by an individual radio frequency chain, including amplifying, down-converting in frequency, and filtering; and, finally, the baseband (or intermediate frequency, IF) signal is sampled and digitized through an analog-to-digital converter (ADC). A central digital signal processor collects the digitized subarray outputs for the sequential processing. In the presence of strong interference signals, AINB needs to be designed and performed across analog subarrays so that low-bit ADC can be used for better power efficiency, as illustrated in Section 1.

Considering the far-field receiving, the subarrays “see” interference signals from the same direction. Thus, the same AINB can be applied for all the subarrays. Letting w collect the beamforming weights of the AINB, we have:(1)$$\begin{array}{c} w={\left[e^{j\phi_{0}},\cdots ,e^{j\phi_{n}},\cdots ,e^{j\phi_{N-1}}\right]}^{T}, \end{array}$$
where $\phi_{n}$ is the value of the phase shifter connected to antenna *n*. As commonly performed in the beamformer design, we divide the angular region of [−90°, 90°] evenly into the following set:(2)$$\begin{array}{c} \Theta =\{-90^{\circ },-90^{\circ }+\delta_{\theta }\cdots ,0^{\circ },\cdots ,90^{\circ }\}, \end{array}$$
where $\delta_{\theta }$ is the minimum angular interval. Then, we design w to make the spatial responses at the discrete angles approach the desired spatial responses. Let $\Theta_{j}$ denote the set of angles in Θ that are closest to interference AoAs and $\Theta_{s}=\Theta /\Theta_{j}$ denote the set difference between Θ and $\Theta_{j}$. For notation clarity, we add subscripts “${\left(\cdot \right)}_{j}$” and “${\left(\cdot \right)}_{s}$” for the variables related to interference and useful signals, respectively.

Steering deep nulls towards the directions of interference signals can be fulfilled by the following optimization:(3)$$\begin{array}{c} f\left(w\right)≊0\;s.t.\;f\left(w\right)=A_{j}^{H}w;\;A_{j}=\left[a_{j}\left({\left[\Theta_{j}\right]}_{0}\right),\cdots ,a_{j}\left({\left[\Theta_{j}\right]}_{L_{j}-1}\right)\right], \end{array}$$
where f(w) collects the beamforming gains achieved by w towards the directions in $\Theta_{j}$, ${\left[\Theta_{j}\right]}_{l_{j}}$ takes the $l_{j}$-th entry in the set $\Theta_{j}$, and $L_{j}$ is the total number of angles in $\Theta_{j}$. The dimensions of zero/one vector and identity matrix can be readily deduced given the context and hence are not explicitly noted in this paper. The steering vector $a_{j}\left({\left[\Theta_{j}\right]}_{l_{j}}\right)$ can be written as:(4)$$\begin{array}{c} a_{j}\left({\left[\Theta_{j}\right]}_{l_{j}}\right)={\left[e^{-jn\pi \sin{\left[\Theta_{j}\right]}_{0}},\cdots ,e^{-jn\pi \sin{\left[\Theta_{j}\right]}_{L_{j}-1}}\right]}^{T}, \end{array}$$
where the antenna spacing *d* has taken half the wavelength to simplify the phase terms. To achieve a flat beam response in the spatial passband, i.e., non-interference directions, we generally minimize the $\ell_{\infty }$-norm of the spatial responses. However, we use the $\ell_{2}$-norm here to make it tractable to jointly achieve the interference suppression and flat beam response in non-interference directions. This will be seen shortly in Section 3.1. In particular, we formulate the following problem:(5)$$\begin{array}{c} \max_{w}\;g\left(w\right)\;\;s.t.\;g\left(w\right)=\frac{1}{L_{s}}{∥A_{s}^{H}w∥}_{2}^{2};\;A_{s}=\left[a_{s}\left({\left[\Theta_{s}\right]}_{0}\right),\cdots ,a_{s}\left({\left[\Theta_{s}\right]}_{L_{s}-1}\right)\right], \end{array}$$
where $L_{s}$ denotes the cardinality of $\Theta_{s}$, and $a_{s}\left({\left[\Theta_{s}\right]}_{l_{s}}\right)$ can be obtained by replacing ${\left[\Theta_{j}\right]}_{l_{j}}$ in (Equation 4) with ${\left[\Theta_{s}\right]}_{l_{s}}$. Combining (Equation 3) and (Equation 5), the AINB design can be formulated into the following optimization problem:(6)$$\begin{array}{c} \min_{w}\;h\left(w\right),\;s.t.\;h\left(w\right)=\frac{\frac{1}{L_{j}}{∥f\left(w\right)∥}_{2}^{2}}{g(w)};\;\left|w\right|=1, \end{array}$$
where the numerator is the normalized power of the spatial responses in $\Theta_{j}$, and |·| takes the point-wise modulus. Due to the constant modulus constraint, problem (Equation 6) is highly non-convex and hence non-trivial to solve efficiently (in real time).

Using $\ell_{2}$-norm in (Equation 5) can greatly simplify the non-convex problem (Equation 6), as will be illustrated in Section 3.1. The simplification further enables us to develop a real-time solver for problem (Equation 6), as will be presented in Section 3.2. Moreover, we unveil in Section 3.3 that initializing the solver properly can help enhance the flatness of the AINB in the non-jamming angular region. The AINB design to be detailed is based on the accurate AoAs of interference signals. Thus, an accurate AoA estimation method will be developed in Section 4.

## 3. AINB Design

In this section, an efficient solver to problem (Equation 6) is proposed under the framework of MM. Specifically, we first simplify the objective function using signal processing techniques/properties, then develop MM-based solver, and finally propose a high-quality initialization for the solver.

### 3.1. Simplifying Beamformer Design Problem

We notice that there is a relationship between the numerator and denominator of the objective function given in (Equation 6). In particular, based on (Equation 3)–(Equation 5), we have:$$\begin{array}{c} {∥f\left(w\right)∥}_{2}^{2}+L_{s}g\left(w\right)={∥A^{H}w∥}_{2}^{2}=\sum_{l=0,\cdots ,L-1}{\left|\frac{1}{N}\sum_{n=0}^{N-1}w_{n}e^{-jn\pi \sin\theta_{l}}\right|}^{2}\\ =\sum_{l=0,\cdots ,L-1}{\left|\sum_{n=0}^{N-1}w_{n}e^{-jnu_{l}}\right|}^{2},\;u_{l}=\pi \sin\theta_{l} \end{array}$$
where $A=[A_{j},A_{s}]$, $w_{n}$ is the *n*-th entry of w given in (Equation 1), L=|Θ| denotes the cardinality of Θ given in (Equation 2), and $\theta_{l}$ is the *l*-th element in Θ. In the final result of the above derivation, $\sum_{n=0}^{N-1}w_{n}e^{-jnu_{l}}$ can be seen as the sampling of the discrete time Fourier transform (DTFT) of w at $u_{l}$. When *L* is sufficiently large, $u_{l}$ approaches a continuous *u* in [−π,π]. Thus, provided a large *L*, we have:(7)$$\begin{array}{c} \left({∥f\left(w\right)∥}_{2}^{2}+L_{s}g\left(w\right)\right)/L≊\frac{1}{2\pi }\underset{\int_{-\pi }^{\pi }du}{\underset{︸}{\lim_{L\to \infty }\frac{2\pi }{L}\sum_{l=0,\cdots ,L-1}}}{\left|\sum_{n=0}^{N-1}w_{n}e^{-jnu_{l}}\right|}^{2}\\ =\frac{1}{2\pi }\int_{-\pi }^{\pi }|\tilde{w}{\left(u\right)|}^{2}du\overset{\left(a\right)}{=}\sum_{n=0}^{N-1}{\left|w_{n}\right|}^{2}\overset{\left(b\right)}{=}N, \end{array}$$
where $\tilde{w}\left(u\right)$ denotes the DTFT of $w_{n}$, the equation $\overset{\left(a\right)}{=}$ is due to the Parseval’s theorem of DTFT [34], and the equation $\overset{\left(b\right)}{=}$ is a result of the constant-modulus constraint on w; see (Equation 6).

The derivation in (Equation 7) further leads to:$$g\left(w\right)=\left(NL-{∥f\left(w\right)∥}_{2}^{2}\right)/L_{s}.$$

Substituting this into (Equation 6), the objective function becomes:$$h\left(w\right)=\frac{\frac{L_{s}}{L_{j}}{∥f\left(w\right)∥}_{2}^{2}}{NL-{∥f\left(w\right)∥}_{2}^{2}},$$
which is a monotonically increasing function of ${∥f\left(w\right)∥}_{2}^{2}$. To this end, h(w) is minimized when ${∥f\left(w\right)∥}_{2}^{2}$ takes the minimum. The above derivation and analysis are formally summarized into the following lemma.

**Lemma** **1.** 
*Provided a large L, the problem (Equation 6) is approximately equivalent to the following simplified minimization:*

(8)
$$\begin{array}{c} \min_{w}{\;∥f\left(w\right)∥}_{2}^{2}=w^{H}R_{j}w,\;s.t.\;R_{j}=A_{j}A_{j}^{H};\;\left|w\right|=1, \end{array}$$

*where $A_{j}$ is given in (Equation 3).*


### 3.2. MM-Based Iterative Solver for Problem (8)

Under the framework of the MM technique, problem (Equation 8) can be iteratively solved by: *majorizing* f(w) at $w_{i}$ (the solution at iteration *i*), leading to the majorization function denoted by $\tilde{f}\left(w,w_{i}\right)$ and *minimizing*
$\tilde{f}\left(w,w_{i}\right)$ subject to the constraint in (Equation 8). We see from (Equation 8) that the objective function is in a quadratic form. According to [35], a quadratic function $x^{H}Lx$ can be majorized at any $x_{0}$ by the following function:(9)$$\begin{array}{c} x^{H}Mx+2ℜ\left\{x_{0}^{H}\left(L-M\right)x\right\}+x_{0}^{H}\left(M-L\right)x_{0}, \end{array}$$
where L and M⪰L are Hermitian matrices. Therefore, to majorize the objective function of (Equation 8), we need to find a Hermitian matrix ${\tilde{R}}_{j}$ such that ${\tilde{R}}_{j}⪰R_{j}$. A straightforward selection of ${\tilde{R}}_{j}$ is $\lambda_{\max}\left\{R_{j}\right\}I$, where $\lambda_{\max}\left\{R_{j}\right\}$ denotes the maximum eigenvalue. Such a selection, however, needs the eigenvalue decomposition of the *N*-dimensional matrix $R_{j}$. This can be avoided applying the following result.

**Lemma** **2.** 
*The maximum eigenvalue of $R_{j}$ is upper bounded by:*

(10)
$$\begin{array}{c} \lambda_{\max}\left\{R_{j}\right\}\leq \bar{\lambda }=NL_{j}, \end{array}$$

*where $L_{j}$ is the cardinality of Θ given in (Equation 3).*


**Proof.** The maximum eigenvalue of a Hermitian matrix can be expressed as the following Rayleigh quotient [Theorem 7.16] [36]:
(11)$$\begin{array}{c} \lambda_{\max}\left\{R_{j}\right\}=\max_{{∥x∥}_{2}^{2}=1}\frac{x^{H}R_{j}x}{x^{H}x}. \end{array}$$Substituting $R_{j}=A_{j}A_{j}^{H}$ into the above definition, we have:
(12)$$\begin{array}{c} \lambda_{\max}\left\{R_{j}\right\}=\max_{{∥x∥}_{2}^{2}=1}\sum_{\begin{array}{c} l_{j}=0,\\ \cdots ,L_{j}-1 \end{array}}{\left|x^{H}a_{j}\left({\left[\Theta_{j}\right]}_{l_{j}}\right)\right|}^{2}\\ \leq \sum_{\begin{array}{c} l_{j}=0,\\ \cdots ,L_{j}-1 \end{array}}\left(x^{H}x\right)\times \left(a_{j}^{H}\left({\left[\Theta_{j}\right]}_{l_{j}}\right)a_{j}\left({\left[\Theta_{j}\right]}_{l_{j}}\right)\right)=NL_{j}, \end{array}$$
where “≤” is based on the Cauchy–Schwartz inequality ([36], Sec. 1.4).    □

Substituting x=w, $M=R_{j}=A_{j}A_{j}^{H}$, $x_{0}=w_{i}$ and $L=\bar{\lambda }I$ into (Equation 9), the objective function of (Equation 8) is majorized at a known point $w_{i}\in C^{N\times 1}$ by the following function:(13)$$\begin{array}{c} \tilde{f}\left(w,w_{i}\right)=-2ℜ\left\{r_{i}^{H}w\right\},\;s.t.\;r_{i}={\left(\bar{\lambda }I-A_{j}A_{j}^{H}\right)}^{H}w_{i} \end{array}$$
where $\bar{\lambda }$ is given in (Equation 10), and the terms independent of w are dropped for brevity. Taking $w_{i}$ as the solution to problem (Equation 8) in the *i*-th iteration, $\tilde{f}\left(w,w_{i}\right)$ is minimized when the real-taking operator achieves the maximum. Due to the constant modulus constraint on w given in (Equation 8), the maximum is achieved when the phases of w are aligned with those of $r_{i}$ in a pointwise manner. To this end, we achieve the following efficient solver for problem (Equation 8).

**Proposition** **1.** 
*The iterative solver to problem (Equation 8) is given by:*

(14)
$$\begin{array}{c} w_{i+1}=e^{j\arg\{r_{i}\}},\;s.t.\;r_{i}={\left(\frac{L_{j}}{N}I-A_{j}A_{j}^{H}\right)}^{H}w_{i}, \end{array}$$

*where $w_{i}$ is the solution in iteration i, $e^{\left(\cdot \right)}$, and arg{·} are pointwise operators, and $\bar{\lambda }$ is derived in Lemma 2.*


Note that the iterative solver has a low computational complexity in each iteration. In particular, the complexity is in the order of $O(N^{2})$, which is dominated by computing $r_{i}$ given in (Equation 14). This owes to the simplification of the objective function achieved in Lemma 1 and the construction of the majorization based on the upper bound derived in Lemma 2.

### 3.3. Initializing $w_{0}$

Given the non-convex nature of problem (Equation 8), the iterative solution achieved in Proposition 1 tends to converge to a local minimum. Namely, the convergence performance of (Equation 14) is closely related to the initialization, i.e., $w_{0}$. Below, we first investigate the impact of $w_{0}$ on the quality of AINB that is measured by the null depth and beam flatness in the spatial passband, and accordingly design a high-quality $w_{0}$. The investigation is performed through a set of simulations.

Figure 2 observes the value of the objective function given in (Equation 8) under the iterative solution given in (Equation 14), where N=16, $\delta_{\theta }=0.1^{\circ }$, $\Theta_{j}=\left\{30.5^{\circ },60.9^{\circ },-50.3^{\circ }\right\}$, and I=800 denotes the total iteration number. Three independent trials are performed, where the initial beamforming weight vector, i.e., $w_{0}$, is independently and randomly generated for each trial. We see that all the three trials have monotonically decreasing value of the objective function before convergence. This validates the effectiveness of the solver developed in Proposition 1. From Figure 2, we also see that the convergence speed is affected by the initialization of $w_{0}$. In particular, the number of iterations required for convergence increases from about 180 to 300 when comparing the second trial with the third one.

Figure 2 also observes the features of the optimized beams in the spatial passband, where $10\log_{10}\left\{w_{i}^{H}R_{s}w_{i}/L_{s}\right\}$ denotes the average power of the beam steered by $w_{i}$ in the spatial passband, while $20\log_{10}\left\{\frac{\max\{|w_{i}^{H}A_{s}|\}}{\min\{|w_{i}^{H}A_{s}|\}}\right\}$ depicts the flatness of the beam, where $R_{s}=A_{s}A_{s}^{H}$ with $A_{s}$ given in (Equation 5). Note that the larger $20\log_{10}\left\{\frac{\max\{|w_{i}^{H}A_{s}|\}}{\min\{|w_{i}^{H}A_{s}|\}}\right\}$ is, the less flat the beam becomes. We see from Figure 2 that the average powers of the optimized beams in the spatial passband are almost identical. This validates the proposed simplification of the objective function for AINB design, as illustrated in Lemma 1. In contrast to the average power, we see from Figure 2 that the flatness of the optimized beams in the spatial passband can substantially vary given different initializations of $w_{0}$. In particular, the difference of the beam flatness is larger than 10 dB between trials 2 and 3.

Figure 3 plots the amplitude responses of the beams steered by $w_{I}$’s obtained from the three independent trials. We see that three deep nulls are produced by the beams obtained in the three trials, and the null depth is below −300 dB. This validates the strong interference suppression ability achieved by the beamformer design problem formulated in Lemma 1 and the iterative solver derived in Proposition 1. We also see from the figure (particular the zoomed-in sub-figure in the middle) that the passband variations in the obtained beams can be subatantially different. This is consistent with what has been observed in Figure 2.

Figure 4 observes the impact of the initialization on the convergence performance of the proposed solver (Equation 14). In particular, the left *y*-axis gives the amplitude variance, i.e., the beam flatness, of the beam steered by $w_{I}$ with respect to that of the beam steered by $w_{0}$, while the converging value of the objective function is shown on the right *y*-axis. We see that similar converging values of the objective function keep steady regardless of the initialized $w_{0}$. We also see that the flatness of the beam steered by the final $w_{I}$ increases linearly overall with that of the beam steered by the initial $w_{0}$.

Given the observations made above, we conclude that *an initialized $w_{0}$ leading to a flat amplitude response in the spatial passband is better in the sense that the optimized beam can also have better flatness in the spatial passband in addition to the strong ability of interference suppression*. On the other hand, we notice that a flat beam under the constant modulus constraint given in (Equation 8) is difficult to realize. The spatial response of the beam steered by w is given by:(15)$$\begin{array}{c} P\left(u_{l}\right)=\sum_{n=0}^{N-1}w_{n}e^{-jnu_{l}},\;l=0,1,\cdots ,L-1, \end{array}$$
where $w_{n}$ is the *n*-th entry of w, $u_{l}=\pi \sin\theta_{l}$, and $\theta_{l}$ is the *l*-th element in Θ given in (Equation 2). Regarding $u_{l}\in \left[-\pi ,\pi \right]$ as the angular frequency, we obtain that $P(u_{l})$ is the (discrete) Fourier transform of $w_{n}$. According to the uncertainty principle of Fourier transform [34], a signal spreading out in one domain corresponds to a signal localized (such as the Dirac delta function) in the dual domain. In our case, the beam flatness implies that $P(u_{l})$ spreads out across $u_{l}$, and hence, $w_{n}$ is expected to be a localized signal sequence. This, however, contradicts with the constant modulus constraint on w, i.e., $|w_{n}|=1\;\forall n$. To steer a beam as flat as possible in the spatial passband, we formulate the following problem to reduce the maximum amplitude variation:(16)$$\begin{array}{c} w_{0}^{*}:{\mathrm{argmin}}_{w_{0}}\frac{\max\{|w_{0}^{H}A|\}}{\min\{|w_{0}^{H}A|\}},\;s.t.\;\left|w_{0}\right|=1, \end{array}$$
where A is the union of $A_{j}$ and $A_{s}$ given in (Equation 3) and (Equation 5), respectively. *Since the initial $w_{0}$ is designed offline*, we can resort to the bio-inspired optimization tools, e.g., the genetic algorithm (GA) and the currently popular grey wolf optimizer [37], etc., to solve the highly non-convex problem (Equation 16).

### 3.4. Speeding up Convergence

As seen from Figure 2, directly solving (Equation 14) can lead to a number of iterations in the order of $10^{2}$. To improve the convergence speed, we employ the widely used squared expectation maximization (SQUAREM) [38] to solve (Equation 14) and summarize the overall iterative processing in Algorithm 1. SQUAREM is a gradient descent method in essence. It performs Cauchy and Barzilai–Borwein (BB) methods in succession in each iteration. Both Cauchy and BB methods perform gradient descent using different step sizes; refer to [38] for details. This leads to the update of w, as enclosed in the projector P(·); see Steps 9 and 12 of Algorithm 1. The BB step size [39] is adopted in SQUAREM, which results in the update of α, as given in Step 8. Updating α uses the intermediate outcomes from Steps 6 and 7. Steps 11 and 12 perform backtracking, a widely used trick in adjusting step size [40], to keep the objective non-increasing. Finally, Algorithm 1 terminates when the condition in either Step 3 or 15 is reached.
**Algorithm 1** MM-based Analog Beamformer Design.1:**Input**: Θ, $\Theta_{j}$, *N* and $w_{0}^{*}$;2:**Initialize** the iteration index i=0 and set ϵ, such as $10^{-6}$, for example; ▹ϵ is used for examining convergence.3:**while**$i\leq I_{\max}$**do**                   ▹$I_{\max}$ is the maximum iteration number.4:    Solving (Equation 14) based on $w_{i}$ gives $w_{i+1}^{\left(1\right)}$;5:    Solving (Equation 14) again based on $w_{i+1}^{\left(1\right)}$ gives $w_{i+1}^{\left(2\right)}$;6:    $v_{1}=w_{i+1}^{\left(1\right)}-w_{i}$;7:    $v_{2}=w_{i+1}^{\left(2\right)}-w_{i+1}^{\left(1\right)}-v_{1}$;8:    $\alpha =-\frac{∥v_{1}∥}{∥v_{2}∥}$;9:    $w=P(w_{i}-2\alpha v_{1}+\alpha^{2}v_{2})$;                    ▹$P\left(x\right)=\frac{x}{\left|x\right|}$ (point-wise).10:    **while** ${∥f\left(w\right)∥}_{2}^{2}>{∥f\left(w_{i}\right)∥}_{2}^{2}$ **do**11:        Update α=(α−1)/2;12:        $w=P(w_{i}-2\alpha v_{1}+\alpha^{2}v_{2})$;13:    **end while**14:    Obtain $w_{i+1}=w$ and i=i+1;15:    Terminate iteration if $∥f\left(w_{i+1}\right){∥}_{2}^{2}-{∥f\left(w_{i}\right)∥}_{2}^{2}<\epsilon$;16:**end while**17:Return $w=w_{i}$.

Algorithm 1 is developed based on the MM framework, which converges to a finite value for the problem such as (Equation 8) with the quadratic objective bounded by zero. Following the convergence analysis in ([41], Section V), one can readily show that solving (Equation 14) iteratively for problem (Equation 8) certainly converges to a stationary point. However, due to the use of SQUAREM, the convergence of Algorithm 1 can only be achieved when the algorithm starts in the vicinity of a stationary point already [42]. As will be shown in Section 5, enabled by the proposed initialization method, Algorithm 1 converges within a few tens of iterations with satisfactory AINBs achieved.

## 4. Estimation of Interference AoAs

As illustrated in Section 3.3, the uncertainty principle determines that the constant-modulus w can only yield a highly localized AINB. This is beneficial in the sense that a deep null can be achieved, as shown in Figure 3. On the other hand, this limits the null width to be as small as possible. The limitation of null width further requires the AoA estimates of the interference signals to be highly accurate. In this section, a two-stage AoA estimation method is developed for the hybrid antenna arrays illustrated in Figure 1. At the core of the method is the conventional ESPRIT. A key problem of applying ESPRIT to the hybrid antenna array is how to design subarray beamforming. This is solved below.

Assume that there are *P* strong interference signals to be suppressed. Denote the *p*-th interference signal as $s_{p}\left(k\right)$ and its AoA as $\theta_{p}$, where *k* denotes the index of discrete time, also known as snapshot. We consider a short time period during which $\theta_{p}\;\forall p$ does not change. The signals impinging on the *N* antennas of subarray *m* can be collected by the following vector:(17)$$\begin{array}{c} x_{m}\left(k\right)=\sum_{p=0}^{P-1}s_{p}\left(k\right)e^{jmNu_{p}}a_{N}\left(u_{p}\right),\;u_{p}=\pi \sin\theta_{p}, \end{array}$$
where $e^{jmNu_{p}}$ is caused by the phase offset between the *m*-th and the first (i.e., reference) subarray, and $a_{N}\left(\theta_{p}\right)$ is an N×1 steering vector. Specifically, $a_{N}\left(\theta_{p}\right)$ is written as:(18)$$\begin{array}{c} a_{N}\left(u_{p}\right)={\left[1,e^{ju_{p}},\cdots ,e^{j\pi \left(N-1\right)u_{p}}\right]}^{T}. \end{array}$$

Let $w_{m}\left(k\right)$ denote the analog beamforming weight vector of subarray *m* at the *k*-th snapshot. The subarray output is the inner product between $w_{m}\left(k\right)$ and $x_{m}$, as given by:(19)$$\begin{array}{c} y_{m}\left(k\right)=\sum_{p=0}^{P-1}s_{p}\left(k\right)e^{jmNu_{p}}w_{m}^{H}\left(k\right)a_{N}\left(u_{p}\right)+\xi_{m}\left(k\right), \end{array}$$
where $\xi_{m}\left(k\right)$ is an AWGN. Stacking the *M* subarray outputs into a vector gives:(20)$$\begin{array}{cc} & \tilde{y}\left(k\right)=\sum_{p=0}^{P-1}\Lambda_{p}W^{H}\left(k\right)a_{N}\left(u_{p}\right)s_{p}\left(k\right)+z\left(k\right),\\ s.t.\; & \Lambda_{p}=\mathrm{diag}\left\{a_{M}\left(Nu_{p}\right)\right\},\;W\left(k\right)=\left[\cdots ,w_{m}\left(k\right),\cdots \right];\\ & a_{M}\left(Nu_{p}\right)={\left[\cdots ,e^{jmNu_{p}},\cdots \right]}^{T};\\ & z\left(k\right)={\left[\cdots ,\xi_{m}\left(k\right),\cdots \right]}^{T}, \end{array}$$
where diag{·} generates a diagonal matrix. Provided that the same beamformer is applied across the *M* subarrays over *K* snapshots, i.e.:(21)$$\begin{array}{c} w_{m}\left(k\right)=w_{m^{\prime }}\left(k^{\prime }\right)=w\;\left(\forall m\neq m^{\prime },\;\forall k\neq k^{\prime }\right), \end{array}$$
we can simplify $\tilde{y}\left(k\right)$ into:(22)$$\begin{array}{cc} y(k) & =\sum_{p=0}^{P-1}a_{M}\left(Nu_{p}\right)g_{p}s_{p}\left(k\right)+z\left(k\right)=As\left(k\right)+z\left(k\right)\\ s.t.\; & A=\left[\cdots ,a_{M}\left(Nu_{p}\right),\cdots \right];\;s\left(k\right)={\left[\cdots ,g_{p}s_{p}\left(k\right),\cdots \right]}^{T}, \end{array}$$
where $g_{p}=w^{H}a_{N}\left(u_{p}\right)$ is the subarray beamforming gain on the *p*-th interference signal.

From (Equation 22), we see that ESPRIT is now applicable provided that (C1) M>P and (C2) the *P* interference signals are not coherent. For (C1), we remark that only those interference signals that are strong enough to cause signal clipping are of interest, and the number of such interference signals can be limited. If P>M, we can trade the time-domain degree of freedom (DoF) for the spatial ones, as designed in [25]. As for the second condition (C2), we remark that techniques such as spatial smoothing [18] can be employed to de-correlate incident signals, as derived in (Equation 22). Given the above reasons, we proceed to assume that both conditions are satisfied and focus on the selection of w for accurate AoA estimation.

From the signal model derived in (Equation 22), we see that the subarray beamforming introduces a coefficient $g_{p}$ to the *p*-th interference signal. To ensure high SNRs for the AoA estimation, we expect that $g_{p}$, if not enhancing, does not attenuate the incident signal too much. However, in the initial stage, we do not have any a priori information on the AoAs of interference signals. Thus, an omnidirectionally flat beam is suggested for a first-stage AoA estimation. Such a beam has been designed in Section 3.3, where the analog beamformer is initialized for the proposed AINB design. In particular, the beamformer is represented by $w_{0}^{*}$ that is optimized via solving (Equation 16) (offline).

By taking $w=w_{0}^{*}$ in (Equation 21), a first stage of AoA estimation is performed, as summarized in Algorithm 2, where the well-developed ESPRIT is encapsulated as a function based on ([18], Section 9.3). Note that performing ESPRIT based on subarray outputs obtained in (Equation 22) only estimates $Nu_{p}$. According to (Equation 17), $u_{p}$ has the range of [−π,π], given $\theta_{p}\in \left[-90^{\circ },90^{\circ }\right]$. Thus, we have $Nu_{p}\in \left[-N\pi ,N\pi \right]$. Since the angle-taking in Step 25 of Algorithm 2 only returns the angle between ±π. Thus, there is an estimation ambiguity in ${\hat{Nu}}_{p}$, which leads to the following *N* possible estimates of $u_{p}$:(23)$$\begin{array}{c} {\hat{u}}_{p}\left(n\right)=\left({\hat{Nu}}_{p}+2n\pi \right)/N,\;n=0,1,\cdots ,N-1. \end{array}$$
**Algorithm 2** An Accurate Two-Stage AoA Estimation Method.1:**procedure**Stage 1(Initial Estimation)2:    Take $w=w_{0}^{*}$ and perform ${\hat{Nu}}_{p}=$ ESPRIT$\left(w_{0}^{*},y\left(k\right),K\right)$;3:    Extract ${\hat{u}}_{p}\left(n\right)$ from ${\hat{Nu}}_{p}$, as performed in (Equation 23);4:    The final estimate is ${\hat{u}}_{p}^{\left(1\right)}={\hat{u}}_{p}\left(n^{*}\right)$ with $n^{*}$ given in (Equation 24);5:**end procedure**6:**procedure**Stage 2(AoA Refinement)7:    **for all** ${\hat{u}}_{p}^{\left(1\right)}\;\left(p=0,1,\cdots \right)$ **do**8:        Take $w=a_{N}\left({\hat{u}}_{p}^{\left(1\right)}\right)$; see (Equation 18) and (Equation 21);9:        Perform ${\hat{Nu}}_{p}^{\left(2\right)}=$ ESPRIT$\left(a_{N}\left({\hat{u}}_{p}^{\left(1\right)}\right),y\left(k\right),K\right)$;10:        Get ${\hat{u}}_{p}^{\left(2\right)}\left(n\right)\;\left(n=0,\cdots ,N-1\right)$ from ${\hat{Nu}}_{p}^{\left(2\right)}$; see (Equation 23);11:        The final estimate is: ${\hat{u}}_{p}=\min_{{\hat{u}}_{p}^{\left(2\right)}\left(n\right)}\left|{\hat{u}}_{p}^{\left(2\right)}\left(n\right)-{\hat{u}}_{p}^{\left(1\right)}\right|$.12:    **end for**13:**end procedure**14:**function**${\hat{Nu}}_{p}=$ ESPRIT$\left(w,y(k),K\right)$15:    Perform subarray beamforming using w, as performed in (Equation 21);16:    Compute $R\left(=\frac{1}{K}\sum_{k=0}^{K-1}y\left(k\right)y^{H}\left(k\right)\right)$; see (Equation 22) for y(k);17:    Estimate the number of paths based on R, yielding $\hat{P}$;18:    The EVD of R gives $R\lambda_{i}=v_{i}\lambda_{i}$ with $\lambda_{i}>\lambda_{i+1}$;19:    Construct $U_{s}=\left[v_{0},\cdots ,v_{\hat{P}-1}\right]$;20:    Denote $U_{1}={\left[U_{s}\right]}_{0:M-2,:}$ and $U_{2}={\left[U_{s}\right]}_{2:M-1,:}$;21:    Compute $C=\left[\begin{array}{c} U_{1}^{H}\\ U_{2}^{H} \end{array}\right]\left[U_{1},U_{2}\right]$.22:    The EVD of C leads to $C=\left[\begin{array}{c} V_{11}\;V_{12}\\ V_{21}\;V_{22} \end{array}\right]\Lambda \left[\begin{array}{c} V_{11}^{H}\;V_{12}^{H}\\ V_{21}^{H}\;V_{22}^{H} \end{array}\right]$;23:    Compute $\Phi =-V_{12}V_{22}^{-1}$;24:    Find the eigenvalues of Φ: $\lambda_{p}\;\left(p=0,\cdots ,\hat{P}-1\right)$;25:    Return ${\hat{Nu}}_{p}=\arg\left\{\lambda_{p}\right\}$.                ▹arg{} takes angle.26:**end function**

This ambiguity can be suppressed by enumerating the possible estimates and identifying the one leading to the largest receiving power as the final estimate [25]. In particular, the following problem is formulated for removing the ambiguity:(24)$$\begin{array}{c} n^{*}:\max_{n}\frac{\sum_{k=0}^{\tilde{K}}{\left|y_{m}\left(k\right)\right|}^{2}}{\tilde{K}},\;s.t.\;w_{m}\left(n\right)=a_{N}\left({\hat{u}}_{p}\left(n\right)\right), \end{array}$$
where $y_{m}\left(k\right)$ is obtained by plugging the above $w_{m}\left(n\right)$ into (Equation 19), while $w_{m}\left(n\right)$ is obtained by taking $u_{p}={\hat{u}}_{p}\left(n\right)$ in (Equation 18). For ease of exposition, we use the *m*-th subarray in (Equation 24) to illustrate the idea of removing the estimation ambiguity in (Equation 23). Given *M* subarrays, we can simultaneously test *M* estimates. Being straightforward, the details are suppressed. Let the final estimate of the first stage be denoted by ${\hat{u}}_{p}^{\left(1\right)}$. As performed in Steps 7–12 of Algorithm 2, the subarrays point towards each of ${\hat{u}}_{p}^{\left(1\right)}$’s, and ESPRIT is performed again with the a greater beamforming gain exploited for better estimation performance, as compared with the first stage.

Next, we remark that the computational complexity of Algorithm 2 is in the order of $O\left(KM^{2}+8{\left(M-1\right)}^{3}\right)$, which is generally low due to the small value of *M*. Note that the first part $O\left(KM^{2}\right)$ is for computing R in Step 16. Normally, we do not include this part in evaluating the complexity of ESPRIT. Here, due to the small value of *M*, $O\left(KM^{2}\right)$ can be comparable to or even larger than the second part, i.e., $O\left(8{\left(M-1\right)}^{3}\right)$. This second part is for the EVD in Step 22 and is an upper limit obtained based on the maximum value of $\hat{P}\left(=M-1\right)$. The complexity of the other EVD in Step 18 is in the order of $O(M^{3})$, which is generally lower than $O\left(8{\left(M-1\right)}^{3}\right)$ for small *M*’s. Thus, the later is used in the overall computational complexity.

## 5. Simulation Results

In this section, the simulation results are provided to validate the proposed designs. Unless otherwise specified, the following parameters are used: N=16, M=4, K=30, $u_{p}∼U_{\left[-\pi ,\pi \right]}$, $s_{p}∼CN\left(0,\sigma_{i}^{2}\right)$, and $\sigma_{n}^{2}=0.01$ Watt, where $s_{p}$ denotes the *p*-th (p=0,1,⋯,P−1) interference signal, as given in (Equation 17), and $\sigma_{n}^{2}$ is the power of the AWGN in the subarray output, i.e., $\xi_{m}\left(k\right)$ given in (Equation 19). When applicable, $N_{\mathrm{trial}}=2\times 10^{4}$ independent trials are performed to obtain an averaged performance. When designing AINB, the whole angular region given in (Equation 2) is considered, and the angle step $\delta_{\theta }$ takes $0.1^{\circ }$. Moreover, to evaluate the proposed AoA estimation method, the state-of-the-art hybrid ESPRIT (H-ESPRIT) [25] is simulated for comparison. Note that H-ESPRIT runs Stage 1 of Algorithm 2 by taking $w=w_{r}$, where $w_{r}$ denotes the subarray beamforming vector with randomly generated phases. The random subarray beamforming used by H-ESPRIT can handle the case that the number of subarrays is smaller than that of incident sources, while the proposed algorithm cannot. However, from the open literature, H-ESPRIT can be most related to the proposed Algorithm 2 in the sense that they are both for hybrid antenna arrays and can estimate the whole angular region. Thus, we employ H-ESPRIT as a benchmark.

For reproducibility, we provide in (Equation 25) the omnidirectionally flat beams for N=8,16 and 24, respectively.

These beams are obtained by solving problem (Equation 16) using the GA toolbox in MATLAB [43]. They are used in the following simulations. Figure 5 plots the amplitude response of the beams steered by $w_{0}^{*}$ given above, in comparison with the beams steered by $w_{r}$ with randomly generated phases in [0,2π]. We see that the optimized $w_{0}^{*}$ has substantially better beam flatness than $w_{r}$.


(25)
$$\begin{array}{c} w_{0}^{*}=\exp\left[j\left(0.2988,1.1812,2.6686,1.3361,0.0484,\right.\right.\left.\left.-3.0455,-0.5214,3.1145\right)\right];\\ w_{0}^{*}=\exp\left[j\left(2.9280,-1.4330,-2.7477,0.3015,-0.4253,\right.\right.2.1168,0.9070,1.5538,-1.3083,1.5321,1.8232,\cdots \\ \left.\left.1.3972,-0.0852,0.4551,-2.0871,-2.5000\right)\right];\;\mathrm{and}\\ w_{0}^{*}=\exp\left[j\left(2.4481,-2.6471,2.2376,-1.6242,-2.4315,\right.\right.-2.7266,-0.5203,-2.6060,-1.1905,3.0591,\cdots \\ 0.5462,-1.8140,2.7608,1.9835,1.6974,-1.5860,0.3268,-1.8279,-1.2872,1.1163,\cdots \\ \left.\left.2.4170,-2.5172,-0.5398,-1.0528\right)\right]. \end{array}$$


Figure 6 plots the MSE of AoA estimates as the INR, the ratio between $\sigma_{i}^{2}$ and $\sigma_{n}^{2}$, increases. A single source is considered in this simulation to focus on illustrating the benefits of using a flat beam for subarrays and adding a second stage of AoA refinement. We see that the AoA estimates from the first stage of the proposed Algorithm 2, using $w_{0}^{*}$ given in (Equation 25), improve over H-ESPRIT [25]. In particular, to achieve the same estimation accuracy, the proposed method reduces the SNR requirement over 10 dB, compared with H-ESPRIT. This manifests the importance of steering an omnidirectionally flat beam for the accurate AoA estimation. We also see from the figure that the AoA estimates from the second stage are further improved over those from the first stage. This is owed to the greater beamforming gain achieved by the directional beam used in the second stage of Algorithm 2.

Figure 7 plots the CDF of the interference suppression after applying the proposed AINB at a subarray. The AINB is designed by running the proposed Algorithm 1, where the AoA estimates obtained for plotting Figure 6 are used for constructing $\Theta_{j}$ required by the algorithm. We see that using the proposed AoA estimation method achieves better interference suppression, as compared with using H-ESPRIT [25]. Take $\sigma_{i}^{2}=-10$ dB for instance. More than 95% of the independent trials suppress interference to lower than −30 dB when using the proposed method for AoA estimation, while only less than 80% of the trials can achieve the equivalent interference suppression when using H-ESPRIT.

Next, we consider the case of multiple interference signals. Figure 8 (upper) plots the CDF of the sum of the absolute AoA estimation errors of multiple signals. We see that the proposed Algorithm 2 achieves better estimation performance than H-ESPRIT. For the proposed method, we see that the second-stage AoA estimation substantially outperforms the first stage. In particular, the absolute estimation error after the second stage is almost always smaller than $0.0531^{\circ }$. This is because the mutual interference between different incident signals can be smaller in the second stage that use a directional beam, as compared with the first stage that uses omnidirectional beams illustrated in Figure 5.

Figure 8 (lower) illustrates the interference suppression ability achieved under different sets of AoA estimates from the upper sub-figure. We see that a better AoA estimation results in a stronger interference suppression. This is reasonable, since the estimates used for the upper sub-figure are passed to Algorithm 2 for designing the AINBs. It is worth pointing out that jointly running Algorithms 1 and 2, the interference suppression is always greater than −65 dB. This not only validates the high performance of the proposed AoA estimation and AINB design but also demonstrates the feasibility of using hybrid antenna arrays to counteract strong interference signals. In addition, from Figure 8, we can observe the impact of angle quantization error (which is resembled by the angle estimation error) on interference suppression.

Figure 9 illustrates the amplitude responses of the AINBs designed using Algorithm 1, where the target and array parameters set for Figure 2 are used here, and $10^{3}$ trials are carried out with independently and randomly generated interference signals and noises. From Figure 9 (top), we see that the absolute AoA estimation errors of all the three sources are consistently smaller than 0.05 over $10^{3}$ independent trials. Thus, the AoA estimations do not have dependencies over the trials. However, in each trial, the obtained AoA estimates are used for designing the AINB by performing the proposed Algorithm 1. The spatial amplitude responses of all AINBs are plotted in Figure 9 (bottom). We see that, due to the consistently accurate AoA estimates, the AINBs obtained in the $10^{3}$ trials overlap.

Moreover, we see from Figure 9 (bottom) that the nulls deeper than −200 dB are achieved in the directions of three interference signals in all the $10^{3}$ independent trials. Moreover, we see that the beam flatness in the spatial passband is much better than that achieved in Figure 3. This validates the high quality of the initial solution (for solver (Equation 14)), as designed in Section 3.3. Lastly, given the overlapping beams obtained over $10^{3}$ trials, we conclude that the proposed Algorithm 1 is reliable in terms of the convergence performance.

Figure 10 plots the convergence curves of performing Algorithm 1 when designing AINBs for Figure 7 and Figure 9. From Figure 10a, we see that, for single-source cases, the proposed Algorithm 1 can converge within 10 iterations. Note that the difference among the convergence curves is caused by the randomly changing AoAs over independent trials in Figure 7. In Figure 10b, we see that, when there are 3 interference signals to nullify, Algorithm 1 converges within 15 iterations. We also see that the convergence curves overlap with each other. This is owed to the high accuracy of the AoA estimates illustrated in Figure 9. Combining all the simulation results presented above, the proposed interference suppression scheme, consisting of Algorithms 1 and 2, can quickly and effectively suppress strong interference signals using hybrid antenna arrays.

Figure 11 illustrates the impact of the quantization bit of the phase shifters in analog subarrays on the proposed design, where the same $2\times 10^{4}$ independent trials from Figure 8 are rerun here. In the method development, we employ the continuous phase shifters, while this can only be approximated in practice. For a phase shifter with *B* bits, it can take $2^{B}$ states, i.e., $e^{j\frac{2\pi b}{B}},\;b=0,1,\cdots ,2^{B}-1$. To count for the phase shifter quantization, we replace each entry of $w^{⋆}$, which is the optimal result of Algorithm 1, with the closest value the phase shifter can take. From Figure 11, we see that the number of quantization bits of the phase shifters has a substantial impact on the interference suppression ability. As the number of quantization bits increases, the optimal $w^{⋆}$ can be better approximated, hence also leading to stronger interference suppression. From Figure 11, we also see that, even for the small 5-bit quantization, the proposed design can achieve greater than 38 dB interference suppression for all $2\times 10^{4}$ independent trials (with random interference directions over trials). Though the interference signals may not be thoroughly suppressed, the power difference between interference and information-bearing signals can be substantially reduced. Consequently, AGCs will readjust, allowing the reception of information-bearing signals without being clipped. This further enables the suppression digital-domain interference, such as the null projection reusing the estimated interference AoAs.

## 6. Conclusions and Future Works

In this paper, we develop a versatile solution for suppressing strong interference signals using the hybrid array of subarrays. This is achieved by a real-time solver for designing the phase-only AINB that steers deep nulls towards the interference directions and maintains flat in the spatial passband. This is also accomplished by an accurate AoA estimation method using the ESPRIT algorithm. In particular, to achieve the same estimation accuracy, the proposed method reduces the SNR requirement by more than 10 dB, compared with state-of-the-art H-ESPRIT [25]. However, we would point out that H-ESPRIT has the capability of augmenting signal subspace dimension, while the proposed method does not have this capability. Moreover, simulation results show that, employing a uniform linear array of four subarrays each with 16 antennas, the proposed solution can provide the interference suppression of 65 dB or higher in the presence of three resolvable interference signals with randomly distributed directions. In addition, the interference suppression larger than 200 dB is also observed when deliberately setting the three interference directions apart. Since the hybrid array has been extensively studied in numerous use cases of mmWave and THz communications, our design is promising to counteract strong interference in many scenarios.

In this work, we rely on the uncertainty principle of the Fourier transform and minimize the null depth to approximately maintain the beam flatness in the spatial passband. However, we notice from Figure 9 that the flatness in the spatial passband can be uneven. This may affect some tasks, e.g., radar/radio sensing, that have stringent requirements on signal strength. As a future work, we will consider to impose the equi-ripple constraints on the beam response in the spatial passband. This is in light of the Parks–McClellan filter design [34], suggesting that the equi-ripple constraint can help increase the minimum mainlobe level. However, due to the non-convexity of the design problem, such a constraint may affect the efficiency and convergence speed/performance of a solver.

## Figures and Tables

**Figure 1 sensors-22-02417-f001:**
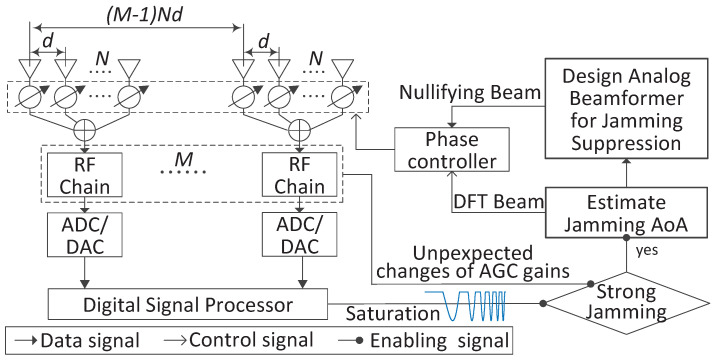
Illustration of the proposed scheme for suppressing strong interference signals based on a uniform linear array with the antenna spacing denoted by *d*. The array is divided into *M* subarrays, each having *N* antennas.

**Figure 2 sensors-22-02417-f002:**
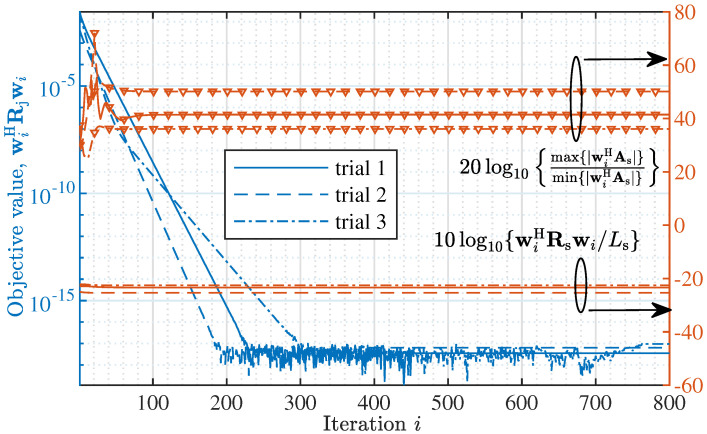
*Left*: The value of the objective function given in (Equation 8) under the iterative solution given in (Equation 14), where N=16, $\delta_{\theta }=0.1^{\circ }$, $\Theta_{j}=\left\{30.5^{\circ },60.9^{\circ },-50.3^{\circ }\right\}$ and I=800 is the total number of iterations. *Right*: Features of the obtained beam in the spatial passband. Three trials are performed with $w_{0}$ randomly and independently generated for each trial.

**Figure 3 sensors-22-02417-f003:**
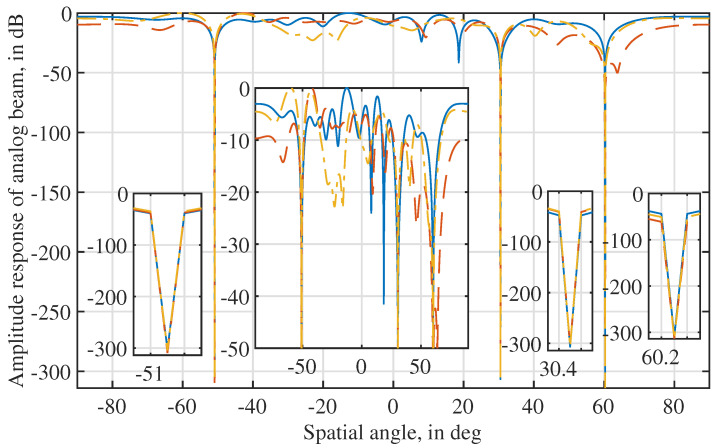
Illustrating the beams steered by the beamformers obtained in Figure 2.

**Figure 4 sensors-22-02417-f004:**
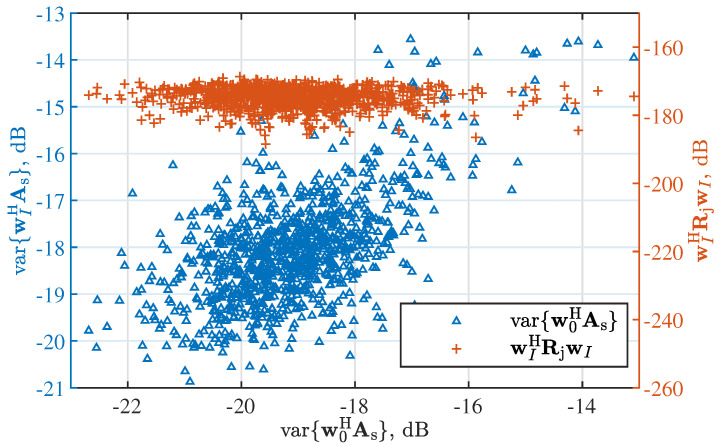
The impact of the initialization on the convergence performance, where the left axis observes the beam flatness and the right one observes the converging value of the objective function in (Equation 8).

**Figure 5 sensors-22-02417-f005:**
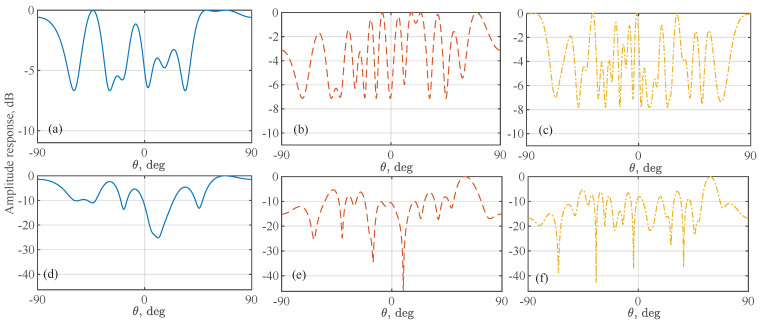
The amplitude response of the beams steered by the analog subarray, where subfigures (**a**–**c**) are obtained using $w_{0}^{*}$ given in (Equation 25), and subfigures (**d**–**f**) correspond to $w_{r}$ with randomly generated phases. The left, middle, and right columns are for N=8, 16, and 24, respectively.

**Figure 6 sensors-22-02417-f006:**
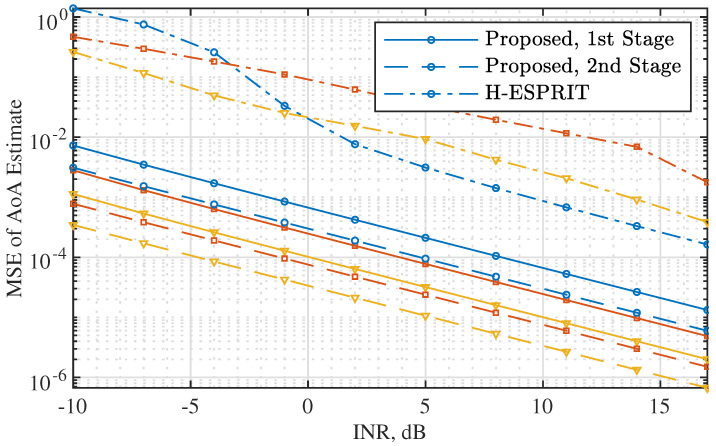
MSE of the AoA estimates versus INR (=$\sigma_{i}^{2}/\sigma_{n}^{2})$, where P=1. The curves with circle, square, and triangle markers are for N=8, 16, and 24, respectively.

**Figure 7 sensors-22-02417-f007:**
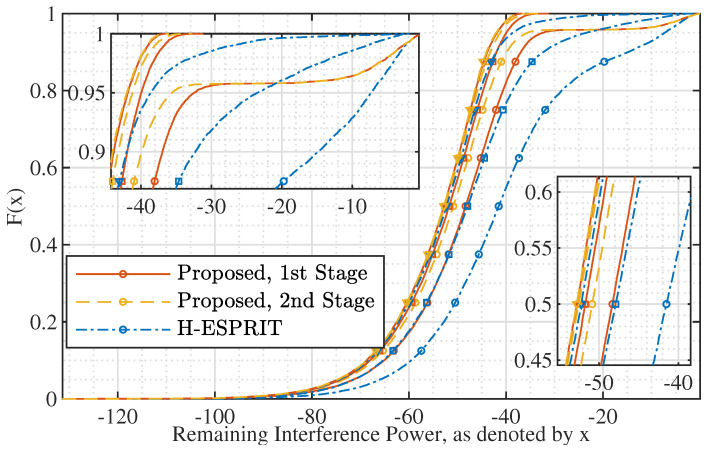
CDF of interference power after subarray beamforming, where N=16. The curves with circle, square, and triangle markers are for $\sigma_{i}^{2}=-10$ dB, −4 dB, and 8 dB, respectively.

**Figure 8 sensors-22-02417-f008:**
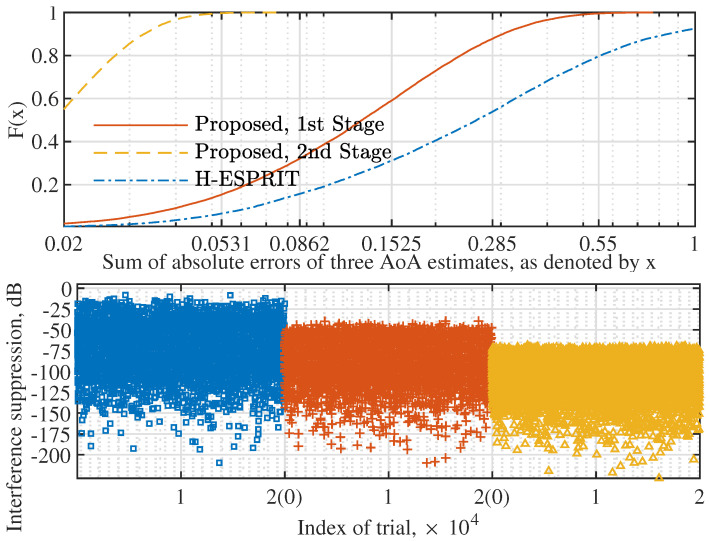
(**Upper**) CDF of the sum of the absolute AoA estimation errors of multiple signals, where N=16, P=3, K=50, and $\sigma_{i}^{2}=0$ dB; (**lower**) illustrating the jamming suppression ability achieved using different sets of AoA estimates, where square, plus, and triangular markers denote H-ESPRIT, the first stage of the proposed method, and the second stage, respectively.

**Figure 9 sensors-22-02417-f009:**
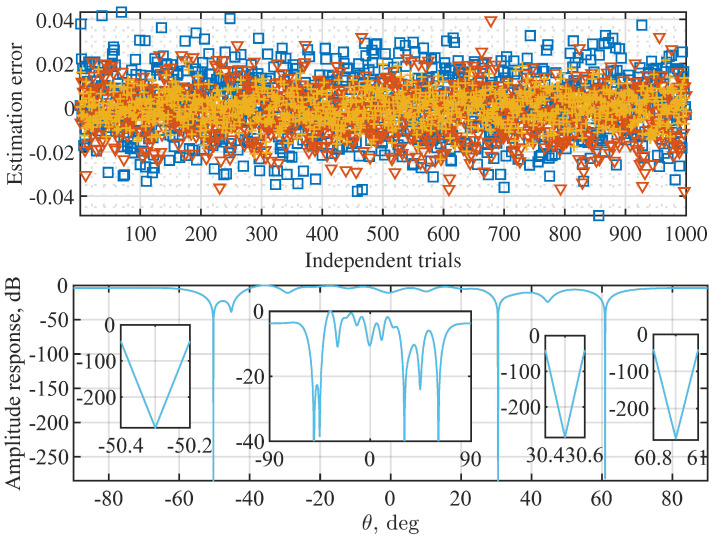
Illustration of the amplitude responses of AINB beams designed by Algorithm 1, where N=16, P=3, K=50, $\sigma_{i}^{2}=0$, and the angles set in Figure 2 are used. The upper figure plots the AoA estimation errors over $10^{3}$ independent trials, where plus, triangle, and square markers are for $30.5^{\circ }$, $60.9^{\circ }$, and $-50.3^{\circ }$, respectively. Note that the second inset from the left in the lower sub-figure is the copy of Figure 9 with the *y*-axis limited to [−40, 0] dB. It is provided to highlight the spatial amplitude response in the region of non-interference directions.

**Figure 10 sensors-22-02417-f010:**
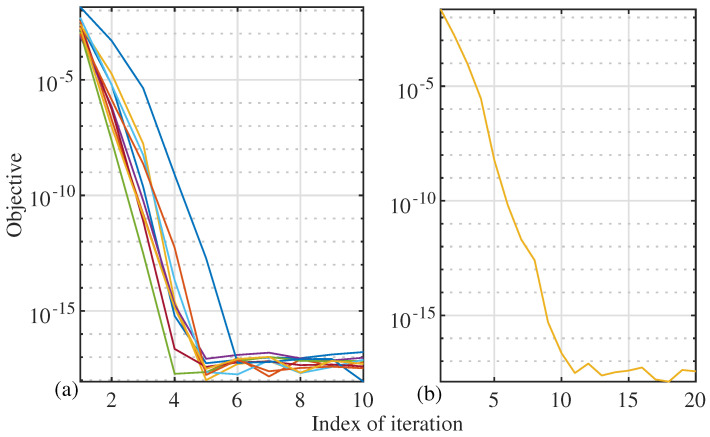
Convergence curves of performing Algorithm 1, where N=16, $\sigma_{i}^{2}=0$ dB, and sub-figures (**a**,**b**) are for the AINB designs in Figure 7 and Figure 9, respectively. Among all the independent trials performed for the figures, 10 trials are randomly selected with their convergence curves presented here.

**Figure 11 sensors-22-02417-f011:**
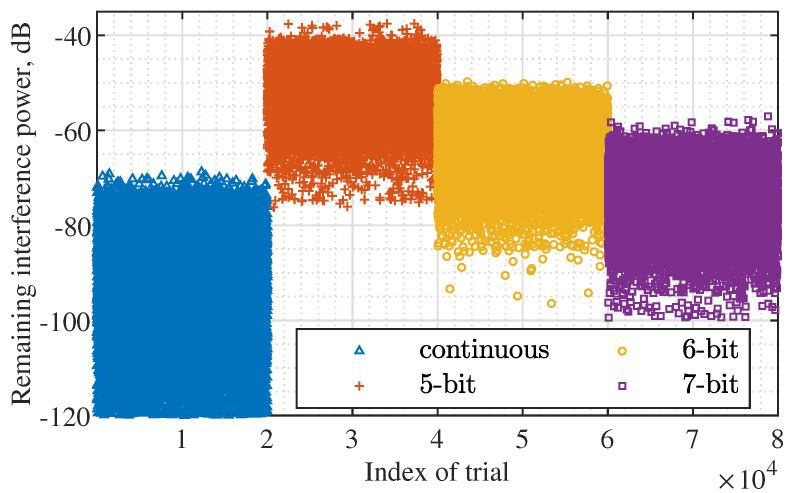
Illustrating the impact of the quantization bit of phase shifters in subarrays on interference suppression.

## Data Availability

Not applicable.

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
