# Peer review of "Analog-Domain Suppression of Strong Interference Using Hybrid Antenna Array"

_sensors, 2022, doi:10.3390/s22062417_

Round 1

Reviewer 1 Report

This paper presents a real-time algorithm to steer nulls towards the interference directions and maintain flat in non-interference directions. In order to realize nulls in the direction of the interference waves, the directions of those estimated with high accuracy. However, there are some unclear points. Please make the following points clear.

Q1

The configuration of the proposed system was introduced in Figure 1. However, the arrangement of the antenna is unclear. If the spacing between the antennas is narrow, electromagnetic coupling between the antennas, that causes distorted directivity, will occur. Please clarify the arrangement of the antenna and the effect of electromagnetic coupling.

Q2

What is the square at the end of sentence in line 179?

Q3

What is the second inset from the left in Figure 9?

Q4

In Figure 9, the nulls deeper than -200 dB are achieved. When the proposed antenna system is used in a real environment, it does not seem to have to be so large because it is affected by thermal noise. On the other hand, I think that a -30 dB null in about -45 degrees will have a bad effect because it is not flat in non-interference directions. Please add your consideration about those.

Q5

In the proposed system, M×N antennas are required and will be extremely large. Please explain the minimum antenna setup required to achieve the desired performance using the proposed method.

Option

The horizontal axis “x” in Figure 7 and Figure 8 (upper) should be written concretely, respectively.

Author Response

Thank you very much for your time and effort in reviewing our paper. Please find our detailed responses to your valuable comments in the uploaded response letter.

Reviewer 2 Report

To greatly reduce the required quantization bits of the analog-to-digital converters and their power consumption. To this goal, the authorship designed a real-time algorithm to steer nulls towards the interference directions and maintain flat in non-interference directions, solely using constant-modulus phase shifters. To ensure sufficient null depth for interference suppression, we also develop a two-stage method for accurately estimating interference directions. The main advantage of proposed solution is that neither training/reference signal nor cooperation/coordination is required. Extensive simulations  show that more than 65 dB suppression can be achieved for three spatially resolvable interference signals yet with random directions. However, there are several major comments the reviewer concerns:

(1) What is the main disadvantage or challenge of the proposes method ?

(2) How to remove the phase ambiguity in the process of estimating DOA due to the hybrid structure

(3) There are crucial literature related to the research topic, which are omitted, cited as follows:

[1] Feng Shu, Yaolu Qin, Tingting Liu, Linqing Gui, Yijin Zhang, Jun Li,
and Zhu Han. Low-Complexity and High-Resolution DOA Estimation for
Hybrid Analog and Digital Massive MIMO Receive Array. IEEE Trans on
Communications, vol.66, no.6, pp.2487-2501, June.2018

[2] D. Hu, Y. Zhang, L. He and J. Wu, "Low-Complexity
Deep-Learning-Based DOA Estimation for Hybrid Massive MIMO Systems With Uniform Circular Arrays," in IEEE Wireless Communications Letters, vol.
9, no. 1, pp. 83-86, Jan. 2020, doi: 10.1109/LWC.2019.2942595.

[3] H. Huang, J. Yang, H. Huang, Y. Song and G. Gui, "Deep Learning for
Super-Resolution Channel Estimation and DOA Estimation Based Massive
MIMO System," in IEEE Transactions on Vehicular Technology, vol. 67, no.
9, pp. 8549-8560, Sept. 2018, doi: 10.1109/TVT.2018.2851783. 

Author Response

Thank you very much for your time and effort in reviewing our paper. We have now addressed all your comments, as detailed in the uploaded response letter.

Reviewer 3 Report

The reviewed article deals with suppressing interferences in wireless networks using hybrid array of subarrays. I consider the topic of the article as very important and timely.

The authors proposed a real-time algorithm to steer nulls towards the interference directions. I think the article is very well written, both in terms of content and language. The authors have correctly defined the problem and given the solution. The theory presented is supported by example simulation results.  

In principle, I have no critical comments, I would only expect a more detailed comparison with the results obtained by other authors in the summary of the final version of the paper.

Author Response

Thank you very much for your time and effort in reviewing our paper. Also, we thank you for the recognition and compliment on our work. In the uploaded response letter, we provide detailed responses to your comments and highlight the text changes made in the revised paper. 

Reviewer 4 Report

[Originality]

In the paper, authors develop a concept of an angle-of-arrival (AoA) estimation by ESPRIT witin a HYBRID (analog/digital) antenna array. Originally, the H-ESPRIT was published in [25] and was validated by the H-MUSIC. The structure of the hybrid array in the submission (Figure 1) and in [25] (Figure 2) is identical, including the phase-only control in the first stage.

Authors claim that novel ideas comprise:

- A simplified objective function (equation 8) being solved iteratively (equation 14);

- A two-stage AoA estimation (an initial estimation + a refinement) by algorithm 2).

Hence, the content of the paper is partially novel.

[Verification]

Performance of the proposed algorithm (one stage, two stages) is compared with the conventional H-ESPRIT. That way, functionality and a higher performance of the presented algorithm is proven by simulations.

[Formal recommendations]

- Figure 5: responses for different N values are difficult to be distinguished.

- Figure 9 top: shown dependencies are hardly understandable. A deeper discussion is welcome.

[Conclusion]

The paper can be accepted for the publication in the current form. Formal improvements can be considered.

Author Response

Thank you very much for your time and effort in reviewing our paper. Your valuable comments have now been addressed, as detailed in the uploaded response letter. 

Reviewer 5 Report

Hybrid beamforming architectures, dynamic analog control of subarrays, phase-only fast and adaptive pattern nulling and angle of arrival estimation techniques have already been studied in the literature intensively. I believe to have the work published in a journal, the key novelties of the paper must be highlighted with fair comparisons from the recent liteature on each part. Otherwise, the work can be considered as a more generic system study for a conference contribution.

Author Response

Thank you very much for your time and effort in reviewing our paper. Please refer to the uploaded response letter for our detailed responses to your comments. 

Round 2

Reviewer 2 Report

The authors have addresses all my concerns. In the current from, it can be published.

Reviewer 5 Report

Thank you for considering my comments and improving the manuscript.